# Application of Skeletonization-Based Method in Solving Inverse Scattering Problems

**Xinhui Zhang [1], Bingyuan Liang [2] and Xiuzhu Ye [1,\***

1    School of Information and Electronics, Beijing Institute of Technology, Beijing 100081, China
2    Institute of Telecommunication and Navigation Satellites, China Academy of Space Technology, Beijing 100094, China
*    Correspondence: xiuzhuye@outlook.com

**Abstract:** In electromagnetic inverse scattering problems, Scattered field commonly needs to be measured by a large number of receiving antennas to provide enough scattered information for image reconstruction, which may increase the cost of the experimental system and require a long testing time. In this paper, a skeletonization-based method was proposed to reduce the number of actual receiving antennas involved in an inverse scattering system. The skeleton points were obtained by performing a strong-rank-revealing QR factorization of Green's function matrix. By measuring the scattered field only at the skeleton points, the number of receiving antennas could be effectively reduced, while the scattered field data at other receiving points could be accurately restored from the skeleton points. The numerical results show that, compared with the frequency domain zero-padding (FDZP) method, the skeletonization-based method was more accurate for antennas distributed in an elliptical shape (such as thorax imaging). In addition, the inverse scattering method using the skeletonization-based method was able to reduce the number of measurements while maintaining an image quality comparable to that of the actual full measurement system. The proposed method can serve as a guidance for building an experimental system for inverse scattering problems, especially for cases when the antennas are elliptically distributed.

**Keywords:** inverse scattering imaging; skeletonization-based method; frequency domain zero-padding (FDZP); back-propagation scheme (BPs); QR factorization; number of actual receiving antennas (NARA)



## 1. Introduction

Electromagnetic inverse scattering problems (ISPs) have been widely studied. In these problems, the aim is to retrieve the distribution of the constitutive properties of unknown scatterers embedded in a domain of interest (DOI) by measuring the scattered field outside the DOI using a certain number of actual receiving antennas (NARA) [1]. Inverse scattering imaging has found applications in non-destructive evaluation [2–4], geological exploration [5,6], and biomedical imaging [7–9]. In particular, in biomedical imaging, the use of ISPs is expected to provide strong support for the diagnosis of breast cancer and detection of stroke due to its super-resolution and quantitative imaging ability [10,11]. The algorithms of inverse scattering imaging can be classified into linear methods and nonlinear methods. Linear methods include the Born approximation algorithm (BA) [12], back-propagation scheme (BPs) [13], and the Rytov approximation method (RA) [14], each of which are suitable for imaging weak scatterers by neglecting the multiple scattering effect. Nonlinear methods include the distorted Born iterative method (DBIM) [15], subspace-based DBIM (S-DBIM) [16], contrast source inversion method (CSI) [17], and subspace-based optimization method (SOM) [18]. Iteration steps are involved in these methods to minimize the cost function constructed by the calculated and measured scattered field.

ISPs are used to determine the constitutive properties of the scatterer by measuring the scattered field. In order to solve ISPs, the scattered field needs to be measured through

a large number of receiving antennas, which may lead to high expenses of the experimental system and may require long testing time. Recently, the compression of near-field sampling points has been studied [19–21]. In reference [22], by using the spectral decomposition of the radiation operator and an approximation of the point spread function, when the field observation domain was on a circular arc, the nonuniform field sampling scheme was proposed to reduce the number of field sampling points for the application of source reconstruction. Compared with the uniform field sampling method, the nonuniform sampling method has a higher source reconstruction accuracy. Furthermore, reference [23] proposed a sampling method to determine the minimum field sampling point when the observation domain was an arbitrary curve by introducing a widely investigated spectrum operator, which can be obtained from evaluating singular values of the radiation/lifting operator. The numerical results showed that the interpolated field matched well with the exact one. The above research either determines the minimum number of sampling points or determines the location of the sampling points, and the field data at other locations need to be further obtained through interpolation methods. In reference [24], a skeletonization-based method was proposed to compress the radiated near-field, which makes use of the low rank characteristics of Green's function matrix. By applying strong rank QR decomposition to Green's function matrix, the skeleton points could be obtained by compressing the rows of Green's function matrix, and the scattered field of other positions of interest could be easily obtained using matrix transformation. Skeletonization-based methods were also applied to accelerate the calculation of method of moments [25,26].

In inverse scattering imaging, the distribution of receiving antennas is commonly distributed in a circle (human brain imaging), while in some specific applications they are distributed conformally into an ellipse, such as in thoracic structure imaging [10]. Therefore, it is necessary to propose a flexible and effective method to reduce the NARA that is suitable for the different distribution shapes of receiving antennas. Inspired by [24], in this article, a skeletonization-based method was proposed to reduce the NARA by setting the receiving antenna at the skeleton points, and the scattered field data at other positions were obtained by using the transformation matrix linked with the scattered field on the skeleton points. This could effectively reduce the NARA required in the inverse scattering imaging system. The main contributions of this paper are as follows:

Firstly, through numerical experiments, we found that increasing the NARA was helpful in improving the imaging results, especially when the scattered field data contained noise.

Secondly, the two interpolation methods used to reduce the NARA, the frequency domain zero-padding (FDZP) method and the skeletonization-based method, were compared in different distribution shapes of the receiving antennas (circular and elliptical). The numerical results showed that compared with FDZP method, the skeletonization-based method achieved a higher interpolation accuracy, especially for the elliptical distribution.

Finally, the skeletonization-based method was applied to reduce the NARA in inverse scattering imaging system with elliptical distribution. The numerical experiments showed that compared with directly increasing the NARA to improve the imaging results, the skeletonization-based method could not only reduce the NARA but also maintain a comparable imaging reconstruction quality under different noise levels.

The structure of this article is as follows. In Section 2, the forward problem theory and skeletonization process are derived. In Section 3, the numerical results are given to verify the effectiveness of the skeletonization-based method to reduce the NARA. Finally, the conclusions are outlined in Section 4.

## 2. Formulation of Forward and Inverse Scattering Problems

### 2.1. Forward Scattering Problem

As shown in Figure 1, we focused on the 2D ISPs under a transverse magnetic wave illumination. The whole system was invariant along the z-axis. The time–harmonic factor $\exp(-i\omega t)$ was adopted. The unknown scatterers were located inside the DOI, $D$, within a free space background with permittivity $\varepsilon_0$, and permeability $\mu_0$. Depending on the

different imaging application requirements, the transmitting and receiving antennas were located around a circle or an ellipse outside the DOI. There were $N_i$ transmitting antennas located at $\mathbf{r}_p$, $p = 1, 2, \ldots, N_i$, generating an incident electric field. For each incidence, the scattered field data was collected by $N_r$ receiving antennas located at $\mathbf{r}_q$, $q = 1, 2, \ldots, N_r$. The collected scattered field data was stored in matrix $\overline{E}^{sca}$ with dimensions $N_r \times N_i$. To calculate the scattered field, the DOI was discretized into $M$ uniform rectangular grids centered at $\mathbf{r}_m$, $m = 1, 2, \ldots, M$. Through the Lippmann–Schwinger equation [27], the total electric field $\overline{E}^{tot}(\mathbf{r})$ in DOI can be written as:

$$\overline{E}^{tot}(\mathbf{r}) = \overline{E}^{inc}(\mathbf{r}) + i\omega\mu_0 \int_D g(\mathbf{r}, \mathbf{r}') \left\{ -i\omega\varepsilon_0 \left[ \varepsilon_r(\mathbf{r}') - 1 \right] \overline{E}^{tot}(\mathbf{r}') \right\} d\mathbf{r}' \quad for \quad \mathbf{r} \in D \quad (1)$$

where $\overline{E}^{inc}(\mathbf{r})$ and $g(\mathbf{r}, \mathbf{r}')$ are the incident electric field in DOI and Green's function in free space, respectively, and $\varepsilon_r$ and $\omega$ are the relative permittivity in the DOI and angular frequency, respectively. Using the pulse basis function and point matching, the Lippmann–Schwinger equation can be discretized into the following matrix equation:

$$\overline{E}^{tot} = \overline{E}^{inc} + \overline{\overline{G}}_D \cdot \overline{\overline{\xi}} \cdot \overline{E}^{tot} \quad (2)$$

where $\overline{\overline{G}}_D$ represents the internal interaction in the scatter, elements in $\overline{\overline{G}}_D$ can be obtained by discretizing the integral operator, and $\overline{\overline{\xi}}(m, m) = \xi(\mathbf{r}_m)$, $m = 1, 2, \ldots, M$ is the contrast of scatter, where $\xi(\mathbf{r}_m)$ is written as:

$$\xi(\mathbf{r}_m) = -i\omega\varepsilon_0 [\varepsilon_r(\mathbf{r}_m) - 1] \quad (3)$$

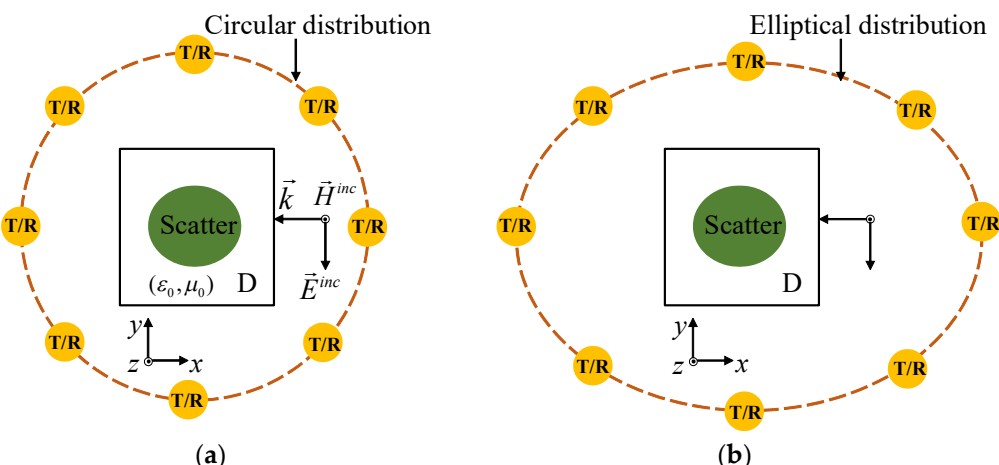

**Figure 1.** Schematic diagram of ISPs and the different distribution shapes: (**a**) elliptical shape; (**b**) circular shape.

The scattered field $\overline{E}^{sca}$ on $N_r$ receiving antennas is written as:

$$\overline{E}^{sca} = \overline{\overline{G}}_S \cdot \overline{J} \quad (4)$$

where $\overline{\overline{G}}_S$ is Green's function matrix denoting the interaction between the induced current $\overline{J} = \overline{\overline{\xi}} \cdot (\overline{\overline{I}} - \overline{\overline{G}}_D \cdot \overline{\overline{\xi}})^{-1} \cdot \overline{E}^{inc}$ in the DOI and the receiving antennas, and $\overline{\overline{I}}$ is the identity matrix. CG-FFT-MOM is used to generate the synthetic measured scattered field data for ISPs [28]. The details of the above process of calculating the scattered field can be found in [1].

### 2.2. Skeletonization Process

In order to reduce the NARA, the positions of the skeleton points need to be obtained. According to the strong-rank-revealing QR factorization [29]:

$$\overline{\overline{P}}^{H}\overline{\overline{G}}_{S} = \overline{\overline{R}}^{H}\overline{\overline{Q}}^{H} \tag{5}$$

where $H$ represents conjugate transpose of a matrix, $\overline{\overline{P}}(\overline{\overline{P}}^{H}\overline{\overline{P}} = \overline{\overline{I}})$ is the permutation matrix, and $\overline{\overline{Q}}$ and $\overline{\overline{R}}$ are the orthogonal matrix containing orthogonal column vectors and the upper triangular matrix (where the value of its main diagonal elements decreases from top to bottom), respectively. The dimensions of matrix $\overline{\overline{G}}_{S}$ and matrix $\overline{\overline{P}}$ are $N_r \times M$ and $N_r \times N_r$, while the dimensions of matrix $\overline{\overline{Q}}$ and $\overline{\overline{R}}$ are $M \times L$ ($L = \min(M, N_r)$) and $L \times N_r$.

If the rank of matrix $\overline{\overline{G}}_{S}$ is $K$, $\overline{\overline{R}}^{H}$ and $\overline{\overline{Q}}^{H}$ can be expressed as:

$$\overline{\overline{Q}}^{H} = \begin{bmatrix} \overline{\overline{Q}}_{11}^{H} & \overline{\overline{Q}}_{12}^{H} \\ \overline{\overline{Q}}_{21}^{H} & \overline{\overline{Q}}_{22}^{H} \end{bmatrix} \tag{6}$$

$$\overline{\overline{R}}^{H} = \begin{bmatrix} \overline{\overline{R}}_{11}^{H} & 0 \\ \overline{\overline{R}}_{21}^{H} & \overline{\overline{R}}_{22}^{H} \end{bmatrix} \tag{7}$$

where the dimensions of $\overline{\overline{Q}}_{11}^{H}$, $\overline{\overline{Q}}_{12}^{H}$, $\overline{\overline{Q}}_{21}^{H}$, and $\overline{\overline{Q}}_{22}^{H}$ are $K \times K$, $K \times (M - K)$, $(L - K) \times K$ and $(L - K) \times (M - K)$, respectively. The dimensions of $\overline{\overline{R}}_{11}^{H}$, $\overline{\overline{R}}_{12}^{H}$, and $\overline{\overline{R}}_{22}^{H}$ are $K \times K$, $(N_r - K) \times K$, and $(N_r - K) \times (L - K)$, respectively. Substituting Equations (6) and (7) into Equation (5), we obtain:

$$\overline{\overline{P}}^{H}\overline{\overline{G}}_{S} = \begin{bmatrix} \overline{\overline{I}}_{K} \\ \overline{\overline{S}} \end{bmatrix}\overline{\overline{G}}_{RS} + \overline{\overline{P}}^{H}\overline{\overline{X}} \tag{8}$$

where $\overline{\overline{G}}_{RS}$, $\overline{\overline{S}}$, and $\overline{\overline{X}}$ can be written as:

$$\overline{\overline{G}}_{RS} = \begin{bmatrix} \overline{\overline{R}}_{11}^{H}\overline{\overline{Q}}_{11}^{H}\overline{\overline{R}}_{11}^{H}\overline{\overline{Q}}_{12}^{H} \end{bmatrix} \tag{9}$$

$$\overline{\overline{S}} = \overline{\overline{R}}_{21}^{H}\left(\overline{\overline{R}}_{11}^{H}\right)^{-1} \tag{10}$$

$$\overline{\overline{X}} = \overline{\overline{P}}\begin{bmatrix} \mathbf{0} & \mathbf{0} \\ \overline{\overline{R}}_{22}^{H}\overline{\overline{Q}}_{21}^{H} & \overline{\overline{R}}_{22}^{H}\overline{\overline{Q}}_{22}^{H} \end{bmatrix} \tag{11}$$

The left and right sides of Equation (8) can be multiplied by the matrix $\overline{\overline{P}}$ at the same time. We can obtain:

$$\overline{\overline{G}}_{S} = \overline{\overline{P}}\begin{bmatrix} \overline{\overline{I}}_{K} \\ \overline{\overline{S}} \end{bmatrix}\overline{\overline{G}}_{RS} + \overline{\overline{X}} \tag{12}$$

The $K + 1$ to $N_r$ singular values of the matrix $\overline{\overline{G}}_{S}$ are ignored because its values are too small, that is, the contribution of $\overline{\overline{X}}$ can be ignored. Then, we can obtain:

$$\overline{\overline{G}}_{S} = \overline{\overline{P}}\begin{bmatrix} \overline{\overline{I}}_{K} \\ \overline{\overline{S}} \end{bmatrix}\overline{\overline{G}}_{RS} \tag{13}$$

$\overline{\overline{G}}_{RS}$ is the row compressed matrix of $\overline{\overline{G}}_S$. The left and right sides of Equation (13) can be multiplied by the matrix $\overline{J}$ at the same time. We then can obtain:

$$\overline{E}^{sca} = \overline{\overline{P}} \begin{bmatrix} \overline{\overline{I}}_K \\ \overline{\overline{S}} \end{bmatrix} \overline{\overline{G}}_{RS} \overline{J} = \overline{\overline{P}} \begin{bmatrix} \overline{\overline{I}}_K \\ \overline{\overline{S}} \end{bmatrix} \overline{E}_S^{SKI} \tag{14}$$

where $\overline{E}_S^{SKI}$ is the scattered field on the skeleton points. It can be seen from Equation (14) that the scattered field in the other positions can be recovered by testing only on the skeleton points.

## 3. Numerical Results

In this section, the numerical results are given to verify the effectiveness of the proposed method. The backpropagation scheme (BPs) was chosen as the inversion algorithm, and the details of which can be found in [1].

As shown in Figure 2, the "Austria" profile was used in the validation of the inverse scattering imaging algorithm, and all the following numerical examples used this model. The "Austria" profile included a ring and two discs, and the relative permittivity of all of its components was 1.1. The ring with an inner diameter of 0.3 m and outer diameter of 0.6 m was centered at (0, −0.2) m, and the centers of the two discs with radii of 0.2 m were located at (−0.3, 0.6) m and (0.3, 0.6) m, respectively. The frequency of the incident waves was 400 MHz. To employ CG-FFT-MOM to calculate the scattering field of the "Austria" profile, the DOI region was discretized into $100 \times 100$ uniform rectangular grids. In the ISPs, the DOI region was discretized into $60 \times 60$ uniform rectangular grids, which was to prevent the inverse crime. Next, we solved the inverse scattering imaging under different noise levels to verify the effectiveness of the proposed method. To show effect of the NARA more clearly, BPs was used for the inversion in all the following examples because there are no parameters involved in this inversion algorithm that are related to noise.

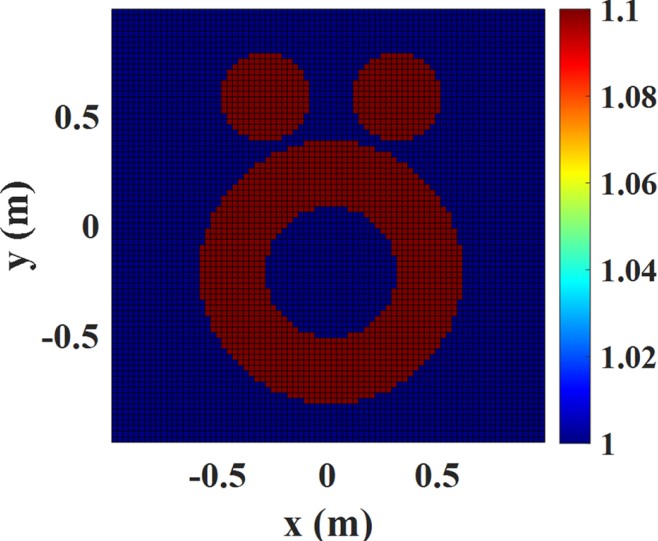

**Figure 2.** Ground truth of the "Austria" profile. The color bar represents the value of the relative permittivity.

To quantitatively evaluate the quality of the reconstructed images, mean-square error (MSE) and structural similarity (SSIM) were adopted. MSE describes the average error of the reconstructed relative permittivity and the original relative permittivity at each grid. SSIM describes the geometric contour similarity between the reconstructed model and the original model.

### 3.1. Two Methods to Reduce the NARA for Distribution Shape

The uniform plane wave illuminated the "Austria" profile, the incident direction of which was the negative *x*-axis direction. A total of 360 receiving antennas, which were equiangularly arranged on a circle (with a radius of 3 m) or an ellipse (with a major axis radius of 3 m, and a minor axis radius of 2.5 m), were used to record the scattered field as a reference to the interpolated ones. Two interpolation schemes to reduce the NARA were adopted:

1.  The linear interpolation method based on frequency domain zero-padding (FDZP) [30]. $N_{FDZP}$ ($N_{FDZP} \ll 360$) receiving antennas were equiangularly placed on a circle, and the scattered field on the 360 receiving antennas could be recovered through the scattered field on the $N_{FDZP}$ receiving antennas using FDZP.
2.  The skeletonization-based method. Implementing the strongrank-revealing QR factorization of Green's function matrix, $N_{SKI}$ ($N_{SKI} \ll 360$) skeleton points were obtained among the 360 position points, which were usually not equiangularly distributed on a circle. The scattered field on the 360 receiving antennas and $N_{SKI}$ skeleton receiving antennas were connected by a transformation matrix. Therefore, it was only necessary to collect the scattered field at $N_{SKI}$ skeleton points.

Here, $N_{FDZP}$ (or $N_{SKI}$) was the NARA. The measured scattered field on the 360 receiving antennas were stored in matrix $\overline{E}_S^M$, while the scattered field on the actual receiving antenna were stored in matrix $\overline{E}_S^A$. In addition, the scattered field $\overline{E}_S^V$ on the 360 virtual receiving antennas could be obtained through matrix $\overline{E}_S^A$ using the two interpolation methods, respectively. To estimate the interpolation error of the two interpolation methods, the following relative error was defined:

$$Err = \frac{\|\overline{E}_S^M - \overline{E}_S^V\|_2}{\|\overline{E}_S^M\|_2}$$

where $\|\ \|_2$ represents the *l*-2 norm.

As shown in Figure 3, the interpolation errors of the two methods were given by changing with the NARA under different distribution shapes of the receiving antennas. From Figure 3, the interpolation error decreased with the increase in the NARA. At the same NARA, the FDZP method had a smaller interpolation error for the circular distribution shape, while the skeletonization-based method had a smaller interpolation error for the elliptical distribution shape. Table 1 shows the required minimum NARA when the interpolation error was less than 1%. The NARA of the skeletonization-based method was 24 for both the circular and elliptical distribution of the receiving antennas. On the contrary, for the FDZP method, the circular distribution required 17 actual receiving antennas, while the elliptical distribution required 28 actual receiving antennas. The FDZP method required a different NARA for the different distribution shapes of receiving antennas, which indicated that the FDZP method was greatly affected by the shape distribution of the receiving antennas.

**Table 1.** The minimum NARA required by FDZP method and skeletonization-based method when the interpolation error was less than 1%.

| | Circular Distribution | | Elliptical Distribution | |
| --- | --- | --- | --- | --- |
| Method | FDZP | SKI | FDZP | SKI |
| Number | 17 | 24 | 28 | 24 |

From the above discussions, to reduce the NARA in the experimental system, different interpolation methods could be selected for the different distribution shapes to sufficiently minimize the NARA. Under the conditions of ensuring the interpolation accuracy, the FDZP method can be adopted for a circular distribution while the skeletonization-based method can be adopted for an elliptical distribution.

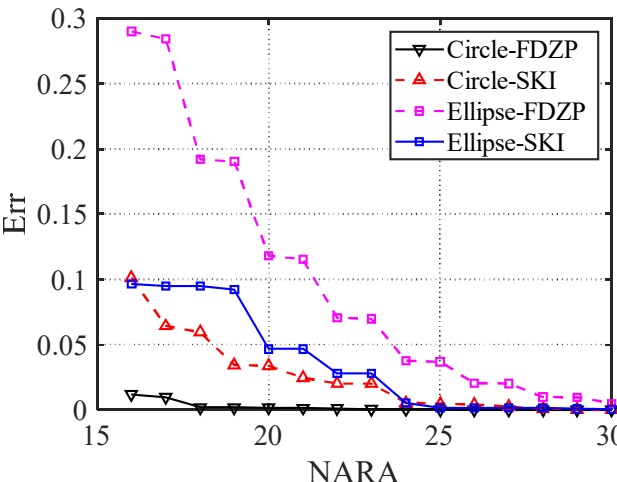

**Figure 3.** Interpolation error using FDZP and skeletonization-based method varying with the NARA for circular and elliptical antenna distribution shapes.

In the inverse scattering imaging system used for a human thorax structure [10], to ensure conformality, the receiving antennas are commonly pasted around the human thorax structure. Since the structure of the human thorax can be approximated by an ellipse, the distribution shape of the receiving antennas is also elliptical. According to the analysis above, in order to reduce the NARA in the imaging system with an elliptical distribution, the skeletonization-based method should be adopted.

### 3.2. Effect of the NARA on the Reconstructed Image

In this section, the effect of the NARA on the inverse scattering imaging with elliptically distributed antennas was studied. All the following numerical examples used an elliptical distribution shape (with a major axis radius of 3 m and a minor axis radius of 2.5 m) of the receiving antennas. Here, the number of transmitting antennas was the same as that of the receiving antennas. The scattered field matrix $\overline{E}_S$ was of dimensions $NARA \times NARA$. The scattered field matrix $\overline{E}_S$ with additive white Gaussian noise was used for the inverse scattering imaging. The additive Gaussian noise level was defined as:

$$nl = \frac{\|\overline{E}_{noi}\|_F}{\|\overline{E}_S\|_F} \times 100\% \tag{15}$$

where $\overline{E}_{noi}$ and $\|\ \|_F$ are the additive Gaussian noise matrix and Frobenius norm, respectively.

Figure 4 shows the changing curves of the MSE and SSIM with the NARA under different noise levels. From Figure 4, we can draw the following conclusions:

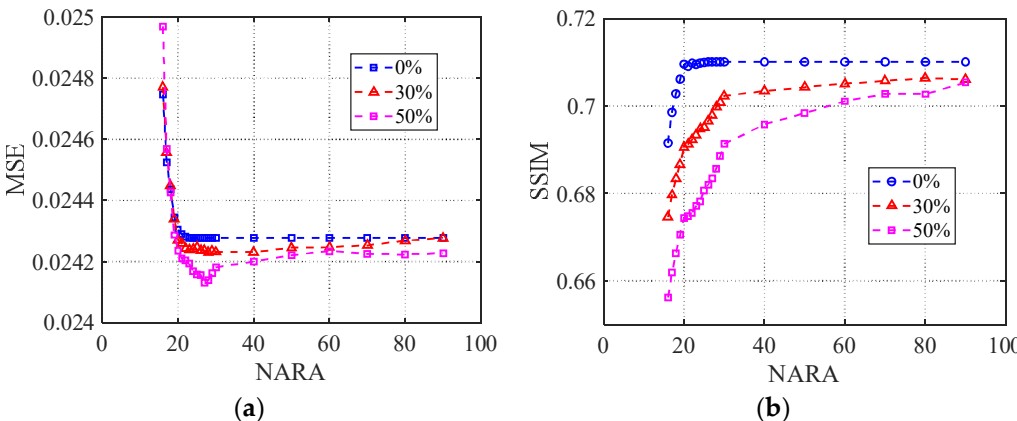

**Figure 4.** Effect of NARA on MSE and SSIM under different noise levels: (**a**) MSE; (**b**) SSIM.

Firstly, in the noise-free case, when the NARA was greater than or equal to 20, the MSE and SSIM did not improve with the increase in the NARA. This showed that in the noise-free case, 20 receiving antennas were already adequate for the inversion.

Secondly, in the noisy cases, the SSIM became larger with the increase in the NARA under same noise level, which showed that increasing the NARA was beneficial for improving the imaging quality when the scattered field contained noise. The larger the noise level was, the more significant the improvement was. For example, when the noise level was 30%, the SSIM was 0.675 for NARA equal to 16, while the SSIM was 0.706 for NARA equal to 90. In addition, when the noise level was 50%, the SSIM was 0.657 for NARA equal to 16, while the SSIM was 0.705 for NARA equal to 90. On the contrary, the increase in the NARA had little effect on the MSE.

Finally, in the noisy case, there were slight changes in the MSE and SSIM curves when the NARA was larger than or equal to 70, which showed that the imaging results could be improved by increasing the NARA with limitations.

According to the above discussions, we can conclude that for a real experimental system with noise, the NARA should be greater than 20. However, in real imaging applications with a limited space (such as thorax imaging), only a limited number of antennas can be arranged, and a large number of antennas will inevitably increase the expense of the experimental system and require a longed testing time. Therefore, one can first measure the scattered field with a small NARA and then interpolate the scattered field to a higher dimension by using the skeletonization-based method.

### 3.3. Reducing NARA by Skeletonization-Based Method

In this session, we proposed an inverse scattering method in cooperation with the skeletonization-based method. Here, we considered the inverse scattering imaging case when the actual receiving antennas were located at the skeleton points on the ellipse. The number of skeleton points was set to 16. That is, the number of actual transmitting antennas and receiving antennas were both 16. The scattered field on the actual receiving antennas was stored in the matrix $\overline{E}_{S\,(16\times16)}$. $\overline{E}_{S\,(16\times16)}$ can be converted to $\overline{E}_{S(NVRA\times16)}^{SKI}$ ($NVRA > 16$) (the number of virtual receiving antennas is abbreviated as NVRA) using the skeletonization-based method. In other words, increasing the number of virtual receiving antennas can allow the obtaining of more scattered field information while keeping the transmitting antenna unchanged. $\overline{E}_{S(NVRA\times16)}^{SKI}$ was used as the input of the inverse problem algorithm for the imaging. Since $\overline{E}_{S(NVRA\times16)}^{SKI}$ contains more scattered field information, the results of the inverse scattering imaging were improved.

Figure 5 shows the changing curves of the MSE and SSIM versus the NVRA. We can conclude that, in the same noise level, compared with the direct use of $\overline{E}_{S\,(16\times16)}$ for inversion imaging, using $\overline{E}_{S(NVRA\times16)}^{SKI}$ could effectively improve the imaging quality. The larger the NVRA value was, the better the imaging quality was (especially the SSIM). Table 2 exhibits the statistical results of the MSE and SSIM using different scattered field matrices for the inversion under different noise levels. Compared with the direct use of $\overline{E}_{S\,(16\times16)}$ for the inverse imaging, the use of matrix $\overline{E}_{S(90\times16)}^{SKI}$ could effectively improve the results of the inverse scattering imaging, and the accuracy of the inversion imaging using $\overline{E}_{S(NVRA\times16)}^{SKI}$ was comparable to that using $\overline{E}_{S(NVRA\times16)}^{Direct}$, which showed that the skeletonization-based method could not only reduce the required NARA but also maintain a high imaging accuracy compared with the direct increase in NARA.

Finally, Figure 6 also shows the results of the inversion imaging using different matrices with 50% noise, which was consistent with the above conclusions. The skeletonization-based method could reduce the NARA for the elliptical antenna distribution while ensuring the imaging accuracy.

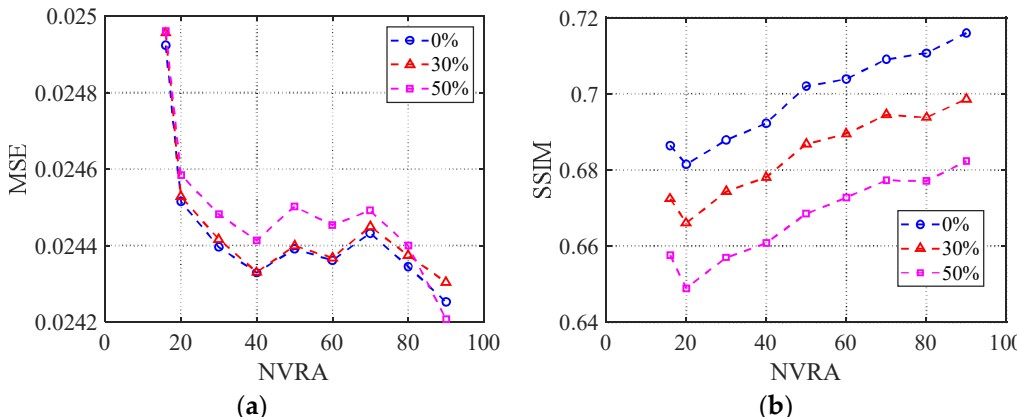

**Figure 5.** Changing curves of MSE and SSIM versus NVRA under different noise levels: (**a**) MSE; (**b**) SSIM.

**Table 2.** Statistical results of MSE and SSIM using different scattered field matrices for inversion under different noise levels.

| Noise Level | 0% | | 30% | | 50% | |
|---|---|---|---|---|---|---|
| Index | MSE | SSIM | MSE | SSIM | MSE | SSIM |
| [1] $\overline{E}_{S(16\times16)}$ | 0.025 | 0.686 | 0.025 | 0.672 | 0.025 | 0.658 |
| [2] $\overline{E}_{S(90\times16)}^{Direct}$ | 0.025 | 0.709 | 0.024 | 0.697 | 0.024 | 0.690 |
| [3] $\overline{E}_{S(90\times16)}^{SKI}$ | 0.024 | 0.716 | 0.024 | 0.698 | 0.024 | 0.682 |

[1] $\overline{E}_{S(16\times16)}$: The number of actual transmitting and receiving antennas was 16. [2] $\overline{E}_{S(90\times16)}^{Direct}$: The number of actual transmitting antennas was 16, and the number of actual receiving antennas was 90. [3] $\overline{E}_{S(90\times16)}^{SKI}$: The number of actual transmitting and receiving antennas was 16, and $\overline{E}_{S(16\times16)}$ was transformed to $\overline{E}_{S(90\times16)}^{SKI}$ by the skeletonization-based method.

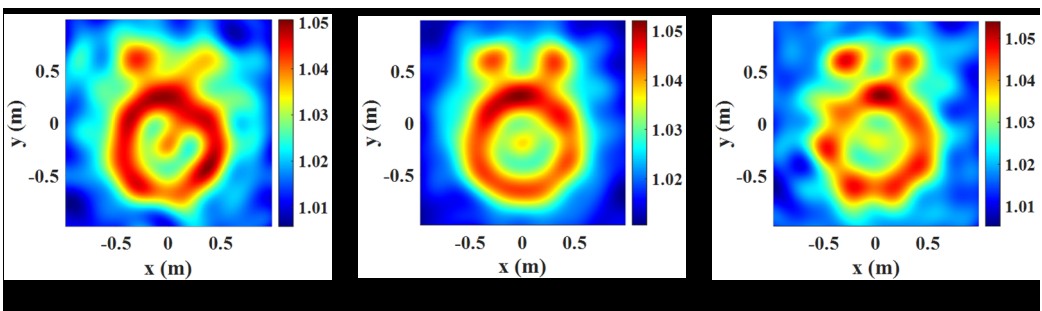

**Figure 6.** Imaging comparison using different scattered field data: (**a**) $\overline{E}_{S\,(16\times16)}$; (**b**) $\overline{E}_{S\,(90\times16)}^{Direct}$; (**c**) $\overline{E}_{S\,(90\times16)}^{SKI}$.

## 4. Conclusions

In this paper, a skeletonization-based method was proposed to reduce the NARA in an inverse scattering imaging system. Through the strong rank-revealing QR decomposition of Green's function matrix, $\overline{G}_S$, skeleton points could be obtained, and the scattered field of other positions of interest could be obtained by the scattered field on the skeleton points, which effectively reduced the NARA. Therefore, more scattered field data on the virtual receiving antenna could be obtained from the scattering field at the skeleton points, which could be used to improve the imaging quality. The numerical results showed that compared with the FDZP method, the skeletonization-based method was more accurate for antennas distributed in an elliptical shape (such as in thorax imaging). The inverse scattering method using the skeletonization-based method could reduce the NARA for an elliptical antenna distribution while ensuring the imaging accuracy. The proposed method can provide

guidance for the layout of receiving antennas in an inverse scattering imaging system, especially for cases when the antennas are elliptically distributed.

**Author Contributions:** Conceptualization, X.Y.; software, X.Z.; validation, X.Z.; writing—original draft preparation, X.Z.; writing—review and editing, X.Y.; supervision, X.Y.; funding acquisition, B.L. All authors have read and agreed to the published version of the manuscript.

**Funding:** This research was funded by the National Natural Science Foundation of China (62001474, 61971036).

**Conflicts of Interest:** The authors declare conflict of interest.

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
