# Peer review of "Application of Skeletonization-Based Method in Solving Inverse Scattering Problems"

_electronics, doi:10.3390/electronics11234005_

Round 1

Reviewer 1 Report

This paper proposes a skeletonization-based method to reduce the number of actual receiving antennas involved in the inverse scattering system. Numerical results show that, compared with the frequency domain zero-padding (FDZP) method, the proposed method is more accurate for antennas distributed in elliptical shape. The paper is in general well written, and the review recommend minor revision. The following questions should be well addressed by the authors:

(1) Does section “2.2. Skeletonization process” solve a linear or nonlinear problem?

(2) What is the advantages and disadvantages of the proposed method over the degree of freedom analysis?

Reviewer 2 Report

The construction and design of the manuscript look good to me. However, here are some concerns for the improvement of the manuscript.

1.      By looking at today’s scenario, everyone is implementing a hybrid concept of working from home and learning from home. To accomplish this, research is going towards MIMO antenna accommodating more number of antennas at the transmitter and receivers to enhance the capacity of the wireless system. With this context, if the number of antennas gets reduced at the receiver side, how the MIMO system will remain efficient for hot-spot-like scenarios? Kindly justify.  

2.      Did the authors consider or design any phantom human model of the head to demonstrate the application of the Skeletonization-Based Method?

3.      How this method will work for Specific absorption rate (SAR) analysis?

Reviewer 3 Report

First, I congratulate the authors on performing an interesting numerical analysis. The article presents skeletonization-based method is proposed to reduce the number of actual receiving antennas involved in the inverse scattering system. The introduction presents the current achievements of other researchers regarding the numerical methods used for this type of research. The authors refer to quite current literature reports. 

Editorial notes:

Table 1, pages 7: There is no reference to the table in the text

Table 2, page 9: There is no reference to the table in the text too.

Figure 6, Page 10: I suggest moving figure 6 to the Numerical Results chapter.

The conclusions was drawn Through the strong rank revealing QR decomposition to the Green's function matrix GS , skeleton points can be obtained, and the scattered field of other positions of interest can be obtained by the scattered field on the skeleton points, which effectively reduces the NARA. The strong point of the article is the extensive numerical analysis of the discussed issue.  As weaknesses, which I suggest in further work, is the lack of experimental research on real objects.

Round 2

Reviewer 2 Report

All comments are addressed.

Author Response

The authors are grateful for the reviewer’s insightful and constructive comments to improve the quality of the manuscript. We have checked the English language for the whole manuscript carefully, and revised accordingly.